# Fostering Culturally Responsive Social-Emotional Learning Practices in Rural Transitional Kindergarten Classrooms

**DOI:** 10.3390/bs15091147

**Published:** 2025-08-23

**Authors:** Xueqin Lin, Josephine Ingram, Chunyan Yang, Rebecca Cheung, Jin Hyung Lim

**Affiliations:** 1Berkeley School of Education, University of California, Berkeley, CA 94720-1670, USA; josephineingram@berkeley.edu (J.I.); rcheung@berkeley.edu (R.C.); jhlim@berkeley.edu (J.H.L.); 2Department of Counseling, Higher Education, and Special Education, University of Maryland, College Park, MD 20742, USA; yangcy@umd.edu

**Keywords:** culturally responsive social emotional learning, early childhood education, rural schools

## Abstract

Despite the positive impact of culturally responsive social emotional learning (CR-SEL) in enhancing students’ academic achievement and emotional resilience, less is known about how it is employed in rural school settings. We employed a case study design to explore how rural transitional kindergarten (TK) teachers in California practice CR-SEL in their classrooms. Ten TK teachers from seven California rural schools were individually interviewed online. Results of the thematic analysis showed three major themes of CR-SEL practices: multicultural and critical perspective development, inclusive environment, and family engagement. Participants identified different strategies to teach CR-SEL practices in their classrooms. Findings extend our understanding of CR-SEL in practice, and provide practical and research implications for school psychologists, educators, and policymakers.

## 1. Introduction

The culturally responsive pedagogy in social-emotional learning (SEL) has been receiving increased attention given its effectiveness in addressing the needs of marginalized and minority students ([38]). While growing evidence highlights the positive impact of culturally responsive SEL (CR-SEL) on student outcomes ([2]; [36]), critical gaps persist in the literature, particularly across geographic and educational contexts, which limit its equitable implementation and scalability in diverse settings. Most existing empirical studies on CR-SEL have centered on urban contexts, neglecting the unique cultural dynamics and resource constraints inherent to rural educational environments ([41]). This urban-centric orientation leaves rural educators without research-informed guidance tailored to their communities’ distinct cultural and logistical realities.

Beyond geographic limitations, existing research on CR-SEL remains constrained by a narrow methodological focus on the quantitative measures, prioritizing program efficacy over implementation processes. This approach provides limited insight into how teachers navigate the complex decisions and adaptations necessary when practicing CR-SEL in authentic classroom settings. Compounding this issue, most purportedly CR-SEL programs lack strong theoretical foundations ([36]), and few meet the inclusion criteria for culturally adapted SEL programs, particularly for racially and ethnically minoritized preschool populations ([2]). The present study addresses the theoretical and empirical gaps by examining CR-SEL practices in rural early childhood education from teachers’ perspectives, grounded in [8]’s ([8]) framework of culturally responsive literacy practices. Using a case study design, we explored how transitional kindergarten (TK) teachers in rural California communities deliver and adapt CR-SEL in their classrooms.

### 1.1. Definition and Importance of CR-SEL

Social-emotional learning (SEL) is defined as a process through which all young people and adults acquire and apply the knowledge, skills, and attitudes to develop five domains of competencies: self-awareness, self-management, social awareness, relationship skills, and responsible decision-making ([14]). Meta-analyses demonstrate the positive relationship between SEL interventions and social, emotional, and academic outcomes in both the short and long term ([11]; [50]). SEL interventions generated positive social-emotional outcomes for young children in rural settings ([50]). In the long term, prosocial skills taught in kindergarten significantly predicted whether students graduated high school on time, completed a college degree, obtained stable employment, and were employed full-time in young adulthood ([30]). Early prosocial skills also had a negative association with living in or being on a waiting list for public housing and receiving public assistance ([30]).

Despite its short- and long-term positive student outcomes, SEL has been critiqued for not meeting the needs of marginalized and minority students ([38]). A growing body of evidence shows the urgent need for CR-SEL, especially in early childhood settings where discriminatory practices are already evident ([44]). [2] ([2]) highlight how biases that lead to the overestimation of challenging behaviors in racial/ethnic minority children often result in disproportionately high rates of exclusionary disciplinary practices against them. These disparities are not merely the product of individual biases but are rooted in structural systems where White, middle-class cultural norms become the standard of acceptable behavior. When SEL practices ignore these structural dynamics and simply promote narrow, culturally non-responsive definitions of social-emotional competencies, they may reinforce deficit views of racial/ethnic minority children and pathologize their culturally grounded behaviors.

CR-SEL integrates the culturally relevant and sustaining pedagogy into SEL programs for diverse and underrepresented students ([38]). It offers transformative possibilities extending beyond cultural inclusion toward educational justice and collective empowerment. Students participating in culturally adapted SEL programs demonstrate substantial improvements across multiple domains ([2]; [36]). These benefits extend to academic achievement, with [41] ([41]) documenting how effective SEL approaches produce not only improved social-emotional outcomes but also improvements in academic and life outcomes. CR-SEL practices recognize how social-historical context shapes definitions of emotional competence across varied communities and cultural contexts and how important it is to integrate cultural context into SEL ([38]). However, most SEL studies have not incorporated theories of culturally responsive pedagogy. A meta-analysis revealed that, among 51 studies reviewed, only five displayed integrations of culturally responsive pedagogy ([41]). This gap exposes a structural oversight within the CR-SEL research.

### 1.2. Conceptual Frameworks of CR-SEL

Researchers have developed frameworks to define the key components of culturally responsive pedagogy ([34]; [45]; [53]). However, few explicitly conceptualize CR-SEL practices in school settings. In fact, even when well-established SEL programs were modified to be responsive to culturally diverse students, they lacked the theoretical underpinnings of cultural adaptation ([36]). Furthermore, existing frameworks that integrate culturally relevant pedagogy into SEL programs focus heavily on culturally responsive classroom activities ([35]), overlooking the need for teachers to work on inter-institutional linkage to deliver CR-SEL, especially in early childhood education, where family-school partnerships are vital for child development. [16] ([16]) demonstrated a case of incorporating SEL and culturally responsive pedagogy approaches into the positive behavioral interventions and support system to transform the school system for CR-SEL practices. The effective CRP practices include the provision of opportunities for students to explore their cultural identities, as well as family engagement, that surveys families’ perceptions of school climate ([16]). These findings highlight the need for teachers not only to design culturally responsive classroom opportunities but also to engage families in a culturally responsive way when delivering SEL practices.

In this study, we employed [8]’s ([8]) framework of culturally responsive literacy practices to guide our understanding of teachers’ CR-SEL practice with young children. [8] ([8]) synthesized from empirical studies the following five interconnected dimensions that embody the foundation of culturally responsive teaching in an early childhood setting: (a) developing a culturally responsive classroom community, (b) family engagement, (c) critical literacy within a social justice framework, (d) multicultural literature, and (e) culturally responsive print-rich environments. This approach establishes what [6] ([6]) describes as learning environments that bridge students’ experiences at home within a school context. Within these spaces, students learn to question whose knowledge counts, how power operates in educational settings, and how to challenge dominant cultural narratives.

Despite key components of culturally responsive pedagogy being proposed in research, how teachers deliver CR-SEL practices in their classrooms varies widely. Findings from prior surveys underscore the need within teacher education for clearer frameworks integrating SEL and CRT, alongside a more unified terminology for these concepts ([51]). [39] ([39]) conducted a survey among 1154 preschool to second-grade teachers to understand their perceptions of SEL integration and their approaches to culturally relevant SEL in classrooms. Findings indicate a range of teacher perceptions on culturally relevant SEL. Some were aware of its implementation (or lack thereof) in their schools, whereas others were unsure if their programs addressed cultural relevance, and a third group dismissed the concept, believing SEL should not be tailored to specific student populations. This highlights the importance of understanding real-life CR-SEL practices to inform teacher practices.

### 1.3. California’s Rural Context and Universal Transitional Kindergarten

Although California is the most populous state, one in 10 children in California reside in rural communities ([5]). Over one-third of its school districts are classified as rural, defined by the state as serving fewer than 600 students and being located more than 25 miles from a city ([29]). Compared with their urban peers, children residing in rural areas experience alarming educational disparities in both mental health and academic achievement. For example, rural children face an increased vulnerability to mental health issues and have less access to mental health services relative to their urban counterparts ([42]). In addition, rural children are more likely to experience a lifetime prevalence of depression and severe behavioral problems compared to urban children ([23]). The lack of accessibility to mental health services, coupled with barriers associated with availability, affordability, and acceptability, is contributing to rural students’ poorer mental health outcomes ([46]). Additionally, an achievement gap exists between rural and nonrural children. For example, rural children were less ready for kindergarten ([31]), and they underperformed nonrural children in both math and reading between third and eighth grade ([28]), possibly due to factors such as resource disparities, teacher retention and support, and access to services and opportunities. These disparities in educational and mental health outcomes underscore the critical need to prioritize the often-overlooked rural youth in California and in the United States.

In July 2021, Governor Newsom signed a 2.7 billion USD bill to expand TK programs across California public schools, establishing the Universal Transitional Kindergarten (UTK) policy ([21]). This initiative aims to provide all four-year-old children in the state with access to high-quality early learning opportunities by the 2025–2026 school year ([10]). TK is a transitional grade level between preschool and kindergarten. Within this evolving landscape, integrating culturally responsive practices into SEL is essential for fostering high-quality learning environments and strengthening teacher-child interactions that support young children’s social-emotional development.

### 1.4. The Present Study

Guided by the five fundamental components of culturally relevant teaching ([8]), this study aims to explore how rural TK teachers employ culturally responsive SEL in their classrooms to promote students’ social-emotional development. Using a case study design, we hoped to address the following question: How do teachers enact SEL in a culturally responsive way in rural TK classrooms?

## 2. Methods

We employed a case study design to address our research question. As a qualitative research approach, a case study design is used to explore and understand complex issues that involve an in-depth examination of a particular case or multiple cases ([18]). We selected the case study design for the following reasons. First, given the purpose of this study is to understand how rural TK teachers enact CR-SEL practices in their classrooms, a case study approach can contribute to deepening our understanding of rural CR-SEL practices and further inform the existing literature and theories related to culturally responsive pedagogy and SEL. Second, the design allows for an in-depth, context-specific examination of real-world classroom practices ([18]), which can inform practices and policy in the UTK implementation to ensure high-quality education.

### 2.1. Participants

Ten TK teachers from seven rural schools in California participated in this study, and the teacher demographic characteristics are presented in Table 1. Pseudonyms are used for all participants to protect their confidentiality. As displayed in Table 1, all the TK teachers in the study were female, with six of them being White. Half of the teacher participants spoke English only, while the other half of them spoke other languages (i.e., Spanish and Korean) as well. Sixty percent of the teachers had over a decade of teaching experience, but most of them were employed at their current sites for less than six years (*n* = 8). Nine out of the ten teachers held at least a bachelor’s degree, with two of them having a master’s degree.

### 2.2. Data Collection Procedure

The study was approved by the Institutional Review Board of the first author’s university in September 2023, prior to data collection. We used convenience sampling by sending out recruitment emails with the study flier through the first author’s professional network in Fall 2023. We recruited ten teacher participants who are TK teachers working in rural California, ideally with previous experience working with other grade-level students. Participants were individually interviewed online for 45–60 min via the HIPAA-compliant Zoom platform between February and June 2024. Participants also completed a brief questionnaire to collect their individual and school demographic information. All participants received a 40 USD gift card to compensate for participation.

In the spring and summer of 2023, we developed the interview protocol, conducted a pilot study to test the protocol with three TK teachers, and modified the protocol accordingly. The interview protocol covered questions that explored participants’ overall experience as a TK teacher, their conceptualization of early childhood SEL, within school collaboration, and family-school partnerships of early childhood SEL. We chose a semi-structured interview, as it offers us both structure and flexibility to collect data. For the present study, we focused exclusively on the interview question about culturally responsive SEL. During the interview, participants were asked the following question: “What does culturally responsive SEL look like in practice for your students?” The first author, who conducted all the interviews, did not initially provide the definition of CR-SEL but gave the clarification when participants raised questions about it. All interview transcripts were automatically transcribed by the Zoom platform.

### 2.3. Data Analysis Procedure

Although all transcripts were automatically transcribed by the Zoom platform, the first author and a post-baccalaureate research assistant validated them to ensure accuracy. All confidential information was eliminated from the transcripts (i.e., child’s name, teacher’s name, school staff’s name, and school’s name) during transcript validation, and pseudonyms were assigned to each participant for confidentiality. The first and second authors of this study used MAXQDA to conduct the interview analysis by “coding the data and collapsing the codes into broad themes” ([17]) relevant to our research focus on CR-SEL. They familiarized themselves with the data following the step-by-step guidelines offered by [12] ([12]). They then took detailed notes and identified broad themes emerging from the data during transcript validation, followed by generating the initial codes to develop the codebook, which were discussed and reviewed. The initial themes were created based on the research questions, guided conceptual framework (i.e., [8]), and the broad codes. Based on the frequency of each theme, initial codes, and the relevance of our research question, we decided to reorganize Bennett’s five dimensions. We merged (a) developing a culturally responsive classroom community with (e) culturally responsive print-rich environments into the theme of an inclusive environment. We also combined (c) critical literacy within a social justice framework and (d) multicultural literature into the theme of developing multicultural and critical perspectives. To ensure the reliability, the first and second authors coded all transcripts independently and discussed resolving any inconsistencies.

### 2.4. Positionality Statement

As researchers with our backgrounds in Education and School Psychology, we are aware of how we bring our perspectives and experiences to this study. The first author received her doctoral training in School Psychology and had years of clinical experience working with schools and communities. The second author is a doctoral student in Language, Literacy, and Culture with previous experience as a K-12 English teacher implementing culturally responsive teaching practices. The third author is a professor in School Psychology and a trained school psychologist with research specialties in risk, resilience, and culture. The fourth author brought her years of working experience in K-12 education as a teacher and administrator and her research in leadership preparation and development. The fifth author is a doctoral student in School Psychology with research and clinical experience in CR-SEL. As outsiders of the participating rural schools, we are aware that our perspectives both enable and constrain understanding of the phenomena under study. Nonetheless, we have been committed to actively interrogating our bias through reflective discussion and acknowledged that our positionality might still introduce an unintentional bias in this study’s design, analysis, and interpretation.

## 3. Results

Results demonstrate three major themes of CR-SEL practices in rural TK classrooms from teachers’ perspectives: multicultural and critical perspective development, inclusive environment, and family engagement. Rural TK teachers varied in their approaches to developing multicultural literacy, largely dependent upon their comfort level. Whereas some teachers explicitly and actively taught it as a part of their SEL curriculum, others reactively addressed the issues when needed. Most teachers actively created an inclusive environment via creating print-rich environments, opportunities, or showing empathy and acknowledgement for culturally related issues. Additionally, several teachers engaged families by culturally communicating with families, providing language support, and inviting families to the classroom for cultural sharing. Table A1 in Appendix A summarizes the concrete list of activities with their quotes that rural TK teachers employ to practice CR-SEL. Figure 1 displays the themes and sub-themes of the findings.

### 3.1. Developing Multicultural and Critical Perspectives

Multicultural and critical perspective development refers to teachers’ approaches to incorporating diverse cultural perspectives into their SEL practices. Our analysis revealed that teachers’ pedagogical strategies fell along a continuum between explicit/active teaching and implicit/reactive teaching, with significant variation in how teachers positioned themselves in relation to cultural difference.

#### 3.1.1. Explicit and Active Teaching

A majority of participating teachers (*n* = 7) explicitly and intentionally incorporated multicultural and critical perspectives into their SEL curriculum. They deliberately selected materials, planned activities, and created lessons to address cultural diversity. These activities include real-life situations and examples (*n* = 4), storytelling (*n* = 2), cultural celebrations (*n* = 2), and self-portraits (*n* = 1). For example, Josephine exemplified her culturally responsive approach, describing her use of real-life examples to teach her students the concept of disability. Josephine observed that one of her students with a physical disability encountered difficulty making friends with others. By explicitly teaching the concept of disability, she enhanced her students’ social awareness to gain empathy with others, which in turn strengthened their interpersonal relationships. Josephine stated, “That class was super accepting of her. I mean, she had a ton of friends. They would get in fights; they would make up”.

Several teachers taught multicultural and critical perspectives via addressing real-world issues such as skin color, religious beliefs, or different family contexts. Margaret encountered a situation where one of her students approached an African American student, asking them why their skin is black. She stepped in and educated her student in Spanish, “We all have different skin colors. We all have different skin tones, but we’re all the same…. It’s just our skin color.“ Margaret used daily living situations to raise her students’ social awareness of racial and ethnic differences in students’ home language, which exemplifies explicit teaching responding to a specific classroom moment. Her choice to conduct this conversation in Spanish also demonstrates her awareness of the intersection between cultural and linguistic responsiveness.

Addressing the same skin color issue, Daisy’s approach targeted self-awareness through her intentional use of diverse literature. At the beginning of the school year, Daisy would usually do a self-portrait activity and read a story called *I Like Myself* to explicitly teach self-acceptance. The story was about an African American little girl feeling embarrassed about going to school because of her “crazy” hair and her mother showing her “all the fun stuff that they can do with their hair”, which made her like her hair. Daisy’s approach represents a deliberate curricular choice connecting self-awareness to racial identity. She helped her students develop positive self-images, as the self-portrait activity connects representation to identity development, inviting children to engage with their own physical characteristics as worthy of artistic representation. Her decision to use this book “at the beginning of the year” positions cultural diversity as foundational rather than supplemental to her classroom community. Daisy does not merely expose students to diversity but actively engages them to consider how self-acceptance connects to physical characteristics associated often with racial identity.

Similarly, Sadie took a direct approach to addressing religious differences, as one of her students was a Jehovah’s witness and she could not celebrate certain holidays that everybody would be talking about. Sadie educated her students that “some people don’t because everybody’s holidays or different events in their life are different”. Her explicit acknowledgment of religious diversity created opportunities for students to develop social awareness across religious boundaries. Unlike approaches that minimize or avoid difference, Sadie’s pedagogy directly engaged with difference as a topic worthy of classroom discussion.

Besides racial, ethnic, and religious differences, Sadie also intentionally educated her class on different family structures, such as two-mom or two-dad families, when some students became curious about others making two holiday gifts for their moms. Said shared, “[Students] are usually really receptive to things once they understand it”. Sadie’s explicit talk about different family structures allowed her students to be more accepting of each other’s differences and to understand different social norms. Through targeted, active teaching methods, rural TK students cultivate both self-awareness and social insight into cultural differences, leading to greater acceptance of themselves and those around them.

#### 3.1.2. Implicit and Reactive Teaching

Half of participating teachers used implicit and reactive teaching strategies, referring to teachers either not taking the initiative to foster multicultural development or their approach emphasizing acceptance over transformation. Several teachers explained that they had to take a cautious and reactive approach, as some multicultural issues can be sensitive in rural and conservative communities. For instance, both Kaia and Josephine worked at Grandview Elementary in Siskiyou County, bordering Oregon, where over 60% of voters supported President Trump in 2024 ([32]). Kaia expressed her caution about teaching sexuality. Because of its sensitivity, Josephine also stated that they need parents’ permission to address gender issues in class.

Despite their caution in teaching students’ multicultural and critical perspectives, the implicit and reactive teaching can also be done effectively. Teacher Samantha took an implicit approach when discussing gender-color preferences:

I feel like they’re so little, we don’t make a big deal of anything like that. As I tell them, my favorite color is blue. I’m a girl, ‘cause they’ll do that a lot, but especially with colors, like you like pink. That’s a girl color, and I’m like, well, what I like is blue.

Rather than explicitly teaching about gender as a social construct, Samantha modeled counter-stereotypical preferences and normalized them through her personal examples. Her approach offered students alternative ways of thinking about gender norms without directly challenging their existing beliefs through formal instruction. This strategy potentially reduces resistance while still expanding students’ thinking about gender.

### 3.2. Creating an Inclusive Environment

Our analysis reveals three distinct but interconnected approaches to creating inclusive environments: creating print-rich environment practices, creating opportunity, and demonstrating empathy through acknowledging differences. Each approach reflects teachers’ strategic adaptation of culturally responsive principles to rural early childhood contexts.

#### 3.2.1. Creating an Inclusive Print-Rich Environment

Three participating teachers described intentional efforts to create an inclusive print-rich environment reflecting diverse cultural backgrounds. It allows young children to foster multiculturalism and social-emotional development. Several examples of creating an inclusive print-rich environment are using different skin colors for characters, having books with a variety of topics, and carefully selecting toys. For example, Daisy described creating representational art materials and explained the rationale for doing so with her students. She intentionally used different skin colors for characters to raise her students’ self-awareness and acceptance of different skin colors. Rather than choosing the traditional flesh-colored crayons, which typically represent only light skin tones, Daisy’s choice of skin color markers helps create material conditions for students to see themselves and others represented authentically.

Like Daisy, Kaia invested time in searching for culturally responsive books and toys for her class. As this year she had several native American students in her class, she carefully picked books representing their Native American culture. She stated, “The people in the books are Native Americans and some of them are cultural kinds of things, and most of them are just books about regular people that look like Native American features so that they can see themselves in that. I just think, having things that are representative of all the kids”.

Not only choosing books that are representative, Kaia was also particular about choosing toys. She said, “I made sure when I was buying dolls for my doll house and my baby dolls, they’re all diverse, print differences, colors, different ages, and different types of people”. Kaia’s deliberate intention to choose books and dolls revealed her strong belief that representation matters across intersecting identities. By incorporating diverse dolls into play materials, Kaia normalized differences within everyday classroom life for her students to develop acceptance of self and others, which is a key aspect of SEL. Through preparing an inclusive, print-rich environment, teachers helped their children to feel seen and foster a sense of belonging at school.

#### 3.2.2. Creating Opportunities

Half of the participating teachers created opportunities and space for their TK students to express who they are and their feelings, reflecting teachers’ active structuring of learning experiences and incorporating cultural diversity. Teachers not only valued the importance for kids to learn via play and social interaction with peers, but they also underscored the need to create space for children to share their personal experiences.

Two teachers created space for students to share “their experience of their life at home” and their ways of expressing different feelings to foster students’ self-management ability. For example, when Samantha was teaching the sad emotion to her students, she gave space for their children to understand and express sadness in their own ways:

One little boy raised his hand and said, “My daddy doesn’t cry.” And I said, “Oh, and you know what? That’s totally okay. Some people don’t cry. That’s okay. It’s okay to cry. But it’s also okay not to cry. We all have different feelings and emotions, and we all go through them differently.” He kinda just nodded his head like, yeah, I come to think of it, I was thinking about that little boy ‘cause that really stuck me. I don’t know if I’ve ever seen him cry.

Samantha gave an example of a little boy observing her father not crying that much and using that to teach her class how different emotions were expressed differently across cultures. She gave the boy space to share his experience and validated him, letting him feel heard when he shared how his family dealt with emotions.

Similarly, Kamila created opportunities for her students to engage in rough play, as she noticed that “rough play [was] extremely common” among her Black students. She said, “And that’s one thing I’m working on changing at my school is, I’d like to have more rough and tumble play allowed, but in a controlled manner”. Kamila dedicated herself to allowing different ways for kids to regulate themselves, which is a key aspect of self-management. These examples showed the importance of teachers creating spaces and opportunities for students to express and regulate their emotions differently.

#### 3.2.3. Showing Empathy and Acknowledgement of Differences 

Four teachers identified showing empathy and acknowledgement of differences as being essential in creating an inclusive environment for teaching CR-SEL practices. By connecting their personal experiences, participating teachers showed an understanding of students’ gender preferences and diverse ways of expressing emotions. Two teachers related to their students’ preference of dressing differently by reflecting on their own personal experiences and allowing students to be themselves without actively interrupting them. For example, Josephine related to her sister’s experience when seeing one of her students bringing his “Elsa backpack from Frozen”. Connecting with her personal experience, she showed empathy and acknowledged differences and let her student express himself freely.

Kamila normalized gender fluidity through personal anecdote and matter-of-fact acceptance to promote students’ self-awareness and self-management. Kamila encountered a student who was “very gender fluid” last year. Kamila reflected on her son’s early childhood experience and realized that she did not care “as long as they’re not hurting anyone”. Her approach frames gender expression as unremarkable and positions acceptance as the default rather than something requiring special attention or explanation. Her reflection created an inclusive environment for diverse gender expressions without potentially drawing unwanted attention to individual students.

Two other teachers acknowledged how emotions were expressed differently across ethnic groups. Samantha explicitly framed emotional expression differences as cultural, and her acknowledgment of gender patterns connected cultural practices to gender socialization, recognizing intersecting influences on children’s emotional development. She “[was] aware of that when she [was] teaching these things”. This suggests her intentional integration of cultural differences when adapting the SEL curriculum, which aligns with a transformative SEL acknowledging how power, privilege, and cultural context shape social-emotional development.

Kamila was aware that having direct eye contact can be difficult for her autistic students and her Hmong students. Thus, when she was teaching SEL, she did not force these students to maintain eye contact. However, she was aware that sometimes it was very difficult because “a lot of times” it was out of habit to say “look at me” to her students. Her acknowledgement of differences allowed her Hmong and autistic students to use different ways to show they were paying attention. These empathy-based approaches demonstrate teachers’ awareness of cultural differences in social-emotional development and their dedication to delivering SEL in a culturally responsive manner.

### 3.3. Promoting Family Engagement

Family engagement was the third essential component of CR-SEL in rural TK classrooms. Participating rural TK teachers overall used three strategies: cultural communication (*n* = 2), culture sharing (*n* = 3), and language support (*n* = 4). These approaches reflect teachers’ strategic navigation of rural community dynamics while attempting to build authentic partnerships with diverse families.

#### 3.3.1. Engaging in Cultural Communication with Families

Cultural communication refers to structured approaches that teachers adopt to build relationships with families from diverse backgrounds through both formal and informal channels, and they explicitly acknowledge and validate families’ diverse cultural, racial, and personal identities in family engagement. Kamila and Merida were two teachers highlighting the importance of cultural communication. Kamila shared that she did home interviews with parents at the beginning of the year to get to know things like holidays that the family celebrates, religious background, and language spoken at home. The home interviews provided her with fundamental knowledge about her students and families that allowed her to be cognizant of cultural differences when engaging with families and students.

Some teachers actively engaged themselves in gaining knowledge about culture. This was evident from Kamila’s example. She shared, “When in doubt, I’ll try to look things up and be like, is this a cultural thing? Or is this just this kid? And sometimes it is a cultural thing. And I’m like, oh, cool, I found something new”. Kamila valued cultural learning as an ongoing inquiry, and her continual learning strengthened her ability to understand her students’ behaviors more.

Merida gave an example of her communicating with families regarding a child attempting to take off his shoes at school and not wearing socks. She was very careful when she tried to ask families to take care of the child’s shoe issue. She reframed potential criticism of a child’s appearance (i.e., oversized shoes) as concern for safety, focusing on concrete impacts rather than abstract norms. In this way, she avoided any miscommunication when addressing students’ behavioral issues at school. Her approach demonstrates strategic communication adapting to cultural and linguistic differences.

#### 3.3.2. Welcoming Culture Sharing

Three teachers, including Ellie, Marget, and Sadie, shared their experience, inviting families to share their cultures to teach CR-SEL. For example, Ellie was the only Asian teacher in their school, where most of the students and teachers were from White or Latino ethnic backgrounds. She fully embraced the importance of integrating culture in teaching SEL. She not only introduced her students to Asian cultural holidays (e.g., Lunar New Year), but also invited families to share their culture to increase a sense of belonging for her students. Ellie shared that one year she had an African American student who did not have any friends at school, and she invited the student’s dad to the classroom to share their culture to foster a sense of belonging for her. This, in turn, has helped the student to feel recognized and like they belong. In her example, culture sharing was an effective strategy to foster students’ peer relationships when they get to know each other not only by their appearance, but also by a deeper understanding of who they are and where they come from.

Ellie also practiced culture sharing by creating multisensory experiences for their students and families. She created multisensory engagement with cultural artifacts spanning from everyday items to artistic expressions. Her approach invites both material and performative cultural sharing from families. By combining material resources with community knowledge, Ellie positioned families as holders of linguistic resources valuable to the classroom, which in turn strengthened students’ and families’ sense of belonging.

Like Ellie, Margaret, and Sadie also regarded culture sharing as an effective way to deliver CR-SEL practices. They incorporated activities related to different cultural holidays (e.g., Black history month). Engaging families in CR-SEL practices makes things “not foreign” and “different”. Instead of teachers finding materials online and explaining cultural holidays themselves, parents bring their own personal experiences and understandings of culture to the classroom, which in turn fosters students’ social awareness.

#### 3.3.3. Providing Language Support

Providing language support refers to specific strategies and resources used to help students and families who speak languages other than English fully participate in the classroom and community. Four participating teachers pointed out the importance of providing language support when engaging families in CR-SEL practices. While Samantha taught Kimochis, an SEL curriculum, to her bilingual students and sent home SEL materials in both English and Spanish, several teachers tried to interact with families in the language they speak when it comes to supporting families’ practice of SEL at home. As shown in Table 1, half of the participating teachers speak a language other than English, and four of them speak Spanish, which aligns with many families’ primary language in their community. For example, both Margaret and Merida stated that their ability to speak Spanish allowed them to respond culturally to the families. Coming from the same cultural background allows them to more easily connect with families and collaborate with them in supporting students’ social-emotional development. As many teachers of this study often informally coach families on SEL practice at home at drop-off or pick-up, it is important that they deliver it in the families’ primary language.

Despite feeling confident in interacting with Latino families, Merida encountered difficulty communicating with families whose primary language was other than English or Spanish. She shared that she had several Thai kids in her class. Not speaking Thai, Merida relied on her parents in the community for translation help when trying to communicate with Thai families. These examples of language support demonstrate teachers’ creative adaptation to linguistic diversity despite limited institutional resources when practicing CR-SEL in their schools. However, most family engagement and language support were largely dependent on teachers’ individual capacities or personal networks.

## 4. Discussion

Findings of this study deepen our understanding of CR-SEL practices in rural contexts and enrich theoretical and empirical literature. Using a case study design, we explored how rural TK teachers in California practice CR-SEL to promote young children’s social-emotional development. The thematic content analysis of interview data revealed three major themes of CR-SEL practices: multicultural and critical perspective development, inclusive environment, and family engagement. Within each theme, we identified a variety of strategies that rural TK teachers used to engage students and families in their CR-SEL practices. These practices reflect both the strengths and challenges of implementing CR-SEL in rural settings, where teacher comfort levels, community norms, and access to professional support shape pedagogical approaches.

### 4.1. Connecting Rural CR-SEL Practices with Conceptual Frameworks

Our findings align closely with [8]’s ([8]) framework on culturally responsive literacy practices in early childhood education. Specifically, “Multicultural and Critical Perspective Development”, connects to Bennett’s components of multicultural literature and critical literacy within a social justice framework. Adding to this domain of the framework, we found that rural TK teachers used both explicit and implicit teaching strategies to promote multicultural competence due to the conservativeness of their rural community. The second theme, “Inclusive Environment”, reflects Bennett’s emphasis on developing a culturally responsive classroom community and culturally responsive print-rich environments. Findings contributed to this framework that constructing an inclusive environment for CR-SEL practice was not just about the materials used (e.g., books, toys, visual displays) but also about relational practices. Teachers created opportunities for students to share their own cultural identities and to learn about others. Teacher-student interactions were shaped by an awareness of positionality, and teachers were intentional about respecting and accepting diverse cultural practices. The third theme, “Family Engagement”, directly aligns with Bennett’s component of the same name. Most importantly, our findings offered specific examples of how to engage families within CR-SEL practices. Teachers communicated with families in ways that valued and validated families’ cultural backgrounds, invited families to share their culture in the classroom, and ensured language support was available when needed. Our findings enriched Bennett et al.’s model by providing specific approaches that teachers employ for CR-SEL in the rural context.

Our findings also indicate that TK teachers incorporated self-awareness, social awareness, self-management, and relationship skills competencies into their CR-SEL practices. For self-awareness, teachers used storybooks on racial identity and self-portrait activities to help children develop a positive self-image and a deeper understanding of their identities. For social awareness, teachers highlighted differences such as physical ability, skin color, and religion to help children recognize and respect diversity. In terms of self-management, teachers focused mainly on embracing culturally diverse ways of expressing and regulating emotions. In promoting relationship skills, teachers used cultural sharing to provide opportunities to learn and interact with peers or families from diverse cultural backgrounds.

However, we found few explicit examples of CR-SEL practices that promoted responsible decision-making. One possible reason is that the five SEL competencies are deeply interrelated; even if not explicitly taught, they are often developed in integrated ways ([49]). Still, the absence of responsive decision-making raises important concerns. Without explicitly supporting this area, we may undermine young children’s voice and agency. As discussed earlier, CR-SEL should not be reduced to simply teaching about culture. Rather, it should involve reconceptualizing student–teacher relationships and constructing inclusive environments that affirm students’ capacity to make their own decisions with appropriate support. Thus, responsible decision-making should also be emphasized in CR-SEL in the rural TK context.

### 4.2. Cultivating Multicultural Perspectives: Explicit and Implicit Approaches

Teachers in this study employed a variety of strategies to cultivate multicultural awareness, ranging from explicit, intentional lessons to more passive, reactive discussions. Some educators, like Josephine and Margaret, directly addressed topics such as disability, race, and family diversity by using real-life examples and student-centered discussions. These findings illustrated how CR-SEL serves to affirm children’s identities while simultaneously advocating for understanding and respect ([41]). However, not all teachers felt equally comfortable leading such discussions. In more conservative rural communities, some educators, like Kaia, avoided explicit conversations about gender or sexuality, fearing backlash from families. Instead, they modeled inclusivity through subtle actions, such as Samantha’s casual remark that her favorite color was blue, even though “the kids think it’s a boy color”. This variation highlights a tension between transformative SEL, which encourages critical discussions of identity and power, and the realities of teaching in communities where such topics may be seen as controversial.

Teachers’ avoidance underscores the complexities educators navigate in rural settings, which frequently present barriers to transformative SEL objectives that seek critical discourse on identity and power dynamics ([20]). While implicit approaches still foster acceptance, they may miss opportunities to help young children critically examine societal norms. First, students lack explicit awareness of cultural dynamics, missing opportunities to develop language for analyzing inequality. Second, silence around certain topics may reinforce dominant narratives by suggesting some differences remain unspeakable. Third, teachers miss empowerment opportunities. Kamila’s dismissive “I don’t care” about gender expression creates safety while foreclosing critical dialogue about identity and social structures.

The developmental appropriateness argument for implicit teaching in early childhood requires scrutiny within rural contexts. While abstract concepts challenge young learners, our findings suggest community surveillance rather than developmental concerns primarily drives implicit pedagogy. Teachers navigate the heightened visibility of rural communities wherein “residents may define their identity, in part, through their connection to their rural place” ([22]), creating conditions whereby challenging local norms risks both professional standing and community belonging.

### 4.3. Importance of Creating an Inclusive Environment for CR-SEL

Our findings reveal teachers’ dedication to creating an inclusive environment via creating a print-rich environment and opportunities and showing empathy and acknowledgement of differences. For creating a print-rich environment, although teachers thoughtfully selected diverse materials (e.g., varied skin-tone crayons, multicultural dolls), no participants described systematically examining materials for bias or stereotypes. This absence possibly reflects structural limitations, such as resource constraints and geographic isolation, deeper than individual oversight. Resource constraints compound pedagogical limitations. Teachers personally purchase culturally responsive materials in rural schools operating with limited fundraising abilities ([33]; [47]). Systemic inequities in school funding, exemplified by Title I allocation formulas that can disadvantage high-poverty rural districts compared to their urban counterparts, compel individual teachers to use their own time and resources to address institutional shortfalls. 

#### The Role of Teacher Self-Reflection and Professional Development

A key factor influencing these pedagogical choices was teacher self-reflection. Educators who connected CR-SEL to their own lived experiences, such as Josephine recalling her sister’s gender-nonconforming childhood, were more likely to teach multicultural topics with confidence. Others, like Kamila, engaged in ongoing cultural learning, researching traditions and family structures to better understand their students. This reflective practice and ongoing cultural learning are essential for CR-SEL practices, as they help teachers recognize their own biases and respond to students’ needs in culturally sustaining ways. However, many rural educators cited limited access to structured professional development in CR-SEL practices, often relying instead on their experiences. This gap in formal training underscores the urgent necessity for context-sensitive professional development tailored to rural educators’ unique challenges, such as balancing conservative community sentiments with the imperative for equitable education ([15]; [48]). Collaborative learning opportunities and peer networks may similarly enhance teachers’ efficacy in implementing CR-SEL strategies, as supported by literature emphasizing the importance of a supportive professional environment ([7]).

### 4.4. Significance of Family Engagement in CR-SEL Practices

Our findings underscore the importance of family engagement in CR-SEL practices, particularly in rural small schools where resources are often constrained. The significance of family components aligns with existing systemic and transformative SEL frameworks ([27]; [40]) and is corroborated by empirical studies highlighting that the effectiveness of SEL curricula is significantly dependent on family involvement ([37]; [43]). Furthermore, findings of this study support existing rural family engagement conceptual frameworks ([13]) that emphasize the necessity of learning about and integrating family cultures, languages, and practices into school environments to effectively engage rural multilingual families.

Most importantly, our findings enrich the existing literature by demonstrating effective strategies for engaging families when practicing CR-SEL, thereby possessing the potential to bridge home-school gaps and facilitate stronger home-school connections. The effective strategies (i.e., cultural communication, culture sharing, and language support) reflect [54]’s ([54]) concept of community cultural wealth, which adopts a strength-oriented approach to perceive cultural knowledge, skills, and abilities that families bring. The effectiveness of these strategies also stems from validating families’ values, reflecting on their strengths, seeking their input, communicating in culturally and linguistically responsive ways, and utilizing knowledge of families’ strengths and backgrounds to support student learning ([13]). When families feel genuinely accepted and included, their willingness to collaborate with schools in SEL practices is markedly enhanced.

It is crucial to note, however, that most language support practices observed in this study relied heavily on teachers’ personal linguistic resources or informal translation arrangements rather than systematic institutional support, a situation likely stemming from pervasive resource constraints in rural communities. While teachers demonstrated commitment to facilitate communication across language differences, their approaches often reflected pragmatic accommodation within these constraints, falling short of transformative practice. This reality underscores the urgent need for systemic investment in dedicated multilingual support services and targeted professional development, enabling a shift from ad hoc individual efforts towards truly transformative and equitable CR-SEL implementation.

## 5. Limitations

Despite the study’s contributions to existing theory and literature, several limitations should be acknowledged. First, the qualitative design with a relatively small sample size limits the generalization of findings to other school contexts and wider populations. Due to the nature of qualitative research, the design of this study is specifically limited to telling the stories of ten teachers in seven rural schools in central California, two years after California approved the UPK Policy and after the COVID-19 pandemic. The data might be different if they were collected at a different time of the year or at different rural schools in the US. However, it is important to state that the purpose of this study is not to generalize findings to a larger population. Instead, it serves to provide space for rural teachers to share their perspectives and experiences of CR-SEL for readers to gain insights from these experiences.

A second limitation is the sampling process. Due to time constraints in data collection, purposive sampling was not successful, and convenience sampling was adopted instead. Although convenience sampling has strengths in terms of accessibility to participants, it also has limitations related to representativeness and sampling bias. The 34 participants who volunteered for this study may not be representative of rural school stakeholders at large. Furthermore, the convenience sampling used in this study involved the snowball strategy, in which we used our professional network to forward recruitment emails to school districts and leaders who might be interested in this study. This might have resulted in the participating rural schools and their leaders being more interested in SEL practices and family-school partnerships than their counterparts who did not volunteer for this study, which could contribute to sampling bias.

## 6. Conclusions and Implications for Practice and Policy

This study is one of the few studies exploring how early childhood teachers deliver CR-SEL practices in rural California schools. Findings from this study provide implications for research and practice, and they also highlight the importance of CR-SEL practices supported at home being congruent with the practices implemented at school.

First, implementing CR-SEL requires not only pedagogical expertise but also significant emotional labor and resilience from educators. Engaging in discussions regarding race and family structures often requires emotional resilience, especially in settings where teachers empathize with students yet anticipate potential community backlash ([20]). Concerns about community reactions to discussions on gender reflect the broader emotional pressures impacting rural educators. To sustain educators’ efforts in these emotionally taxing contexts, it is imperative that schools prioritize teacher well-being through structured supports, such as SEL training targeting educators. For instance, [24] ([24]) discussed how school leaders utilize SEL strategies to help teachers, particularly White educators, manage their emotional responses when engaging in professional learning about race and racism. They highlight that learning about race and racism can trigger defensiveness and shame among White teachers, and SEL strategies can assist in regulating these emotions to support engagement in anti-racist teaching practices. Additionally, establishing peer support networks through practices, such as communities of practice ([52]), professional learning communities ([4]), and teacher discourse communities ([9]), can provide safe forums for teachers to share challenges and strategies, fostering a collaborative approach to addressing sensitive topics in their classrooms ([4]).

Second, the findings in this study indicate that teachers in rural CA schools implemented CR-SEL with few resources, often resorting to personal assets (such as language) or networks, which resulted in a disproportionate emphasis on visual approaches. Thus, significant implications are related to the areas of resource development, preparation, and professional learning, as well as professional standards. With the proliferation of CR-SEL content in the marketplace as well as content available on the internet, teachers need support to both identify and develop truly CR-SEL-aligned resources that can be appropriately curated for the local context. For example, how can a teacher with limited or no fluency in a language other than English be supported to assess the materials (visuals, music, videos, etc.) for appropriateness? Examples of evidence-based effective strategies include engaging in professional learning communities ([3]), utilizing culturally responsive frameworks (e.g., Mirrors, Windows, and Sliding Glass Doors; [26]), activating families’ funds of knowledge ([25]), and leveraging technological and digital resources ([1]). There is high potential for rural teachers to use and integrate artificial intelligence (AI) in CR-SEL practices ([1]). [1] ([1]) revealed that AI integration not only helps teachers to personalize instruction and provide targeted support, but also facilitates collaborative learning experiences, promotes active engagement, and provides real-time feedback to teachers and students. It is important that future research investigate strategies and tools that help teachers navigate challenges in teaching CR-SEL. 

Relatedly, preparation and professional learning can support the implementation of CR-SEL through reflection and community asset identification. As described earlier in this paper, reflective practice and ongoing cultural learning are essential for CR-SEL practices, as they help teachers recognize their own biases and respond to students’ needs in culturally sustaining ways ([19]). These practices and orientations can be introduced and practiced early in preparation so that they are habituated in the early career phase. Further, community asset mapping practices can be incorporated into preparation and professional learning to support teachers with approaches to engage in the local communities they serve so that they can easily identify the individuals, community-based organizations (such as activity centers), and institutions (such as churches) that can serve as key informants and resources in the implementation of CR-SEL.

Finally, the rural context of this study emphasizes how professional standards can support the implementation of CR-SEL. As with many other curricular standards movements, the standards inform the development of materials by teachers and companies, as well as the adoption and selection of content. The findings of this study indicate the need to develop standards that support teachers in reflecting and developing their cultural knowledge. Further, standards can support the curation of appropriate materials as well as the integration of community assets into the selection process. 

## Figures and Tables

**Figure 1 behavsci-15-01147-f001:**
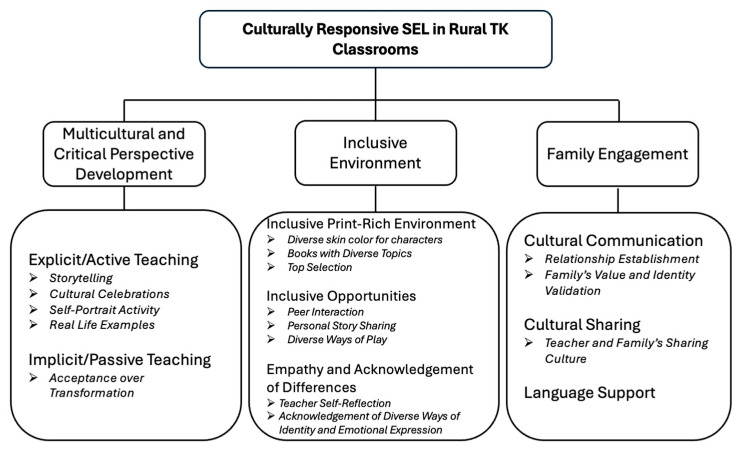
Themes and sub-themes of culturally responsive SEL in rural TK classrooms.

**Table 1 behavsci-15-01147-t001:** Demographic Characteristics of Participating Rural Transitional Kindergarten Teachers.

Participant	Age	Gender	Race/Ethnicity	Language Spoken Other than English	Years of Experience as a Teacher	Years of Experience at Current Site	Highest Degree
Merida	50	Female	Latina	Spanish	24	2	Bachelor’s
Ellie	53	Female	Asian	Korean	15	2	Bachelor’s
Rebecca	36	Female	White	None	6	6	Bachelor’s
Daisy	30	Female	White	None	7	3	Bachelor’s
Sadie	53	Female	White	None	29	22	Bachelor’s
Margaret	49	Female	Latina	Spanish	5	5	Master’s
Josephine	43	Female	White	None	10	10	Bachelor’s
Kaia	56	Female	White	None	30	1	Teaching Credential + 80 units
Kamila	33	Female	White	Spanish	11	2	Bachelor’s
Samantha	29	Female	Latina	Spanish	6	2	Master’s

## Data Availability

Data is unavailable due to privacy or ethical restrictions.

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
