# Peer review of "Fostering Culturally Responsive Social-Emotional Learning Practices in Rural Transitional Kindergarten Classrooms"

_behavsci, 2025, doi:10.3390/bs15091147_

Round 1

Reviewer 1 Report

Comments and Suggestions for Authors

This is a well-argued and structured research paper. The authors have done their due diligence to describe in detail the research methodology and methods, as well as present the findings with relevant supporting participant data. Overall, I was impressed by the quality of this research study. I have no suggestions for improving the quality of this manuscript and recommend accepting as is.

Author Response

Comment 1: This is a well-argued and structured research paper. The authors have done their due diligence to describe in detail the research methodology and methods, as well as present the findings with relevant supporting participant data. Overall, I was impressed by the quality of this research study. I have no suggestions for improving the quality of this manuscript and recommend accepting as is.

Response 1: Thank you very much for spending time reviewing our manuscript. We appreciate your positive comments :) 

Reviewer 2 Report

Comments and Suggestions for Authors

Dear authors,
I read the manuscript with interest because it announces an important and relevant theme for the specialised literature, and you have indeed approached a difficult and interesting subject.
However, I have a few suggestions:
- From the point of view of respecting the indications for authors established by the journal, I would ask you to check the way of indicating the references in the text and at the end of the manuscript.
From the methodological point of view, I noticed that you are not very clear, explicit about the methods used. You opted for a qualitative methodology (case study and semi-structured direct interviews) but you did not describe the protocol followed in the case study, and regarding the interview, you did not describe the themes addressed, how you interpreted the data (frequency of certain themes, their relevance for your research and the objectives you pursued, etc.). It is very important what the professional characteristics of the research participants (professional experience, level of education, etc)
From the point of view of research ethics, you claim that your research and analysis respect personal data (you have anonymised the identification data of the research participants), but the names of the teaching staff appear in the manuscript.
Also, in writing the article, it is very often not clear when we are discussing quotes from interviews and when our opinion as researchers comes into play. Please check.
I wish you success.

Comments on the Quality of English Language

I am not a native English speaker, but as far as my knowledge allows, I believe it can be improved.

Author Response

Dear authors,
I read the manuscript with interest because it announces an important and relevant theme for the specialised literature, and you have indeed approached a difficult and interesting subject.

Response 2.1: Thank you for your feedback. We truly appreciate it.

However, I have a few suggestions:
- From the point of view of respecting the indications for authors established by the journal, I would ask you to check the way of indicating the references in the text and at the end of the manuscript.

Response 2.2: Thank you for your suggestion. We double checked the format requirements of the journal and addressed your comment by using the Behavioral Sciences Microsoft Word template file to format our manuscript.

From the methodological point of view, I noticed that you are not very clear, explicit about the methods used. You opted for a qualitative methodology (case study and semi-structured direct interviews) but you did not describe the protocol followed in the case study, and regarding the interview, you did not describe the themes addressed, how you interpreted the data (frequency of certain themes, their relevance for your research and the objectives you pursued, etc.). It is very important what the professional characteristics of the research participants (professional experience, level of education, etc)
From the point of view of research ethics, you claim that your research and analysis respect personal data (you have anonymised the identification data of the research participants), but the names of the teaching staff appear in the manuscript.

Response 2.3: Thank you very much for your observation! To integrate your feedback, we added information about how we developed and tested our interview protocol and made it clear that the first author conducted all the interviews. We also added the information about the collection of demographic information via a brief questionnaire. We made it clearer that our interviews covered themes like participants’ overall experience as a TK teacher, their conceptualization of early SEL, within school collaboration and family-school partnerships of early SEL. For this study, we focused exclusively on one interview question asked about CR-SEL.

We also appreciated your comment on how we interpreted the data. Thus, we explicitly explained on page 6 that “The initial themes were created based on the research questions, guided conceptual framework (i.e., Bennett et al., 2018) and the broad codes. Based on the frequency of each theme, initial codes and the relevance of our research question, we decided to reorganize Bennett et al's five dimensions. We merged (a) developing a culturally responsive classroom community with (e) culturally responsive print rich environments into the theme of inclusive environment. We also combined (c) critical literacy within a social justice framework and (d) multicultural literature into the theme of developing multicultural and critical perspectives.”

We also agreed that the importance of presenting the professional characteristics of the participants, which were described in the “participants” section. We moved the table of demographic characteristics to this section for readers’ convenience as well. As mentioned in this section, “Pseudonyms are used for all participants to protect their confidentiality”. We reinstated that in the data analysis procedure. To further protect their confidentiality, we now removed the demographic characteristics of schools where the participants worked.

Also, in writing the article, it is very often not clear when we are discussing quotes from interviews and when our opinion as researchers comes into play. Please check.
I wish you success.

Response 2.4: Thank you for pointing it out. We revised our results section accordingly by moving most quotes to Table A1 in Appendix A.

Comments on the Quality of English Language

I am not a native English speaker, but as far as my knowledge allows, I believe it can be improved.

Response 2.5: Thank you for your feedback. We thoroughly reviewed our transcripts by English native speakers to improve the quality of English language. We also used GenAI for superficial text editing (e.g., regarding grammar, spelling, punctuation and formatting).

Reviewer 3 Report

Comments and Suggestions for Authors

CR-SEL is a structure that combines two broad concepts—SEL and cultural responsiveness—and its conceptual scope is excessively wide, making it difficult to apply concretely in practice. Because CR-SEL is so comprehensive and wide-ranging, there is also the issue that researchers and teachers may interpret it differently. Furthermore, since most SEL activities may appear to contain some “CR-SEL-like” elements, the boundary of what constitutes genuine CR-SEL can become blurred.

Therefore, theoretical refinement and practical delineation are necessary. In other words, although many educational activities may appear to include elements of CR-SEL, in reality, they often fail to fully reflect students’ diverse cultural contexts. Ultimately, it is not simply a matter of whether activities exist, but how they are designed and implemented that matters.

Major Comments

  1. Narrow the core scope of CR-SEL
    It is necessary to narrow the core scope of CR-SEL. For example, focusing on a specific domain such as “SEL strategies that enhance cultural responsiveness in teacher–child interactions” would be more effective. It is uncertain whether such a narrowing is feasible within this current manuscript, but at present, the range the paper seeks to cover is far too broad.
  2. Position the theoretical framework more prominently
    It would be clearer if the Bennett framework or CASEL’s five core SEL competencies were explicitly presented at the beginning of the study and if the results were reorganized according to this framework. Currently, the connection between the theoretical framework and the themes is not clearly articulated, making the theoretical component appear weak.

  3. Clarify the theoretical contribution
    Although the study draws on the Bennett et al. (2018) framework, it should clearly indicate what new elements it adds to the framework or what modifications for rural contexts are suggested. Rather than simply applying the Bennett framework, the study should provide implications such as proposing a new model or concept for how CR-SEL should be modified or expanded in rural contexts.

  4. Present the activities (practical content) more concisely
    Reduce the number of interview quotations and instead present the core activities and their meaning through tables and diagrams. The cases should be structurally organized, and the main text should focus on summarized interpretations.
    Alternatively, specific activities or examples could be compiled into a table. Although the manuscript includes many teacher interview excerpts and examples, the activities are scattered throughout the narrative. Table 3 contains some activities, but they are very brief and at a keyword level. Even though the manuscript narratively organizes the activities under the “three major themes” (developing multicultural and critical perspectives, creating inclusive environments, and family engagement), they are not easily viewed at a glance.

    If a concrete list of activities were provided, the practical implications of the findings would be clearer and the usefulness to readers would increase. However, rather than simply listing activities, they should be organized by theme and connected to the theoretical context (CASEL’s five SEL competencies and the CR-SEL framework). We recommend summarizing them in a table or as an appendix rather than expanding the text. Indicating which SEL competencies each activity relates to would strengthen both the theoretical and practical contributions of the study.

  5. Emphasize structural issues in family engagement and language support
    The manuscript should stress that family engagement and language support are largely dependent on teachers’ individual capacities or personal networks. Framing this as a structural problem would give the study a stronger policy message.

Minor Comment

  1. Clarify the concept of TK
    Considering international readers, the concept of TK (Transitional Kindergarten) may be unfamiliar to those outside the United States. Adding one or two sentences to explicitly state that “TK is a transitional grade level between Kindergarten and Preschool” would make the manuscript clearer.

Author Response

Comments and Suggestions for Authors

CR-SEL is a structure that combines two broad concepts—SEL and cultural responsiveness—and its conceptual scope is excessively wide, making it difficult to apply concretely in practice. Because CR-SEL is so comprehensive and wide-ranging, there is also the issue that researchers and teachers may interpret it differently. Furthermore, since most SEL activities may appear to contain some “CR-SEL-like” elements, the boundary of what constitutes genuine CR-SEL can become blurred.

Therefore, theoretical refinement and practical delineation are necessary. In other words, although many educational activities may appear to include elements of CR-SEL, in reality, they often fail to fully reflect students’ diverse cultural contexts. Ultimately, it is not simply a matter of whether activities exist, but how they are designed and implemented that matters.

Response 3.0: Thank you so much for your insights.

Major Comments

  1. Narrow the core scope of CR-SEL
    It is necessary to narrow the core scope of CR-SEL. For example, focusing on a specific domain such as “SEL strategies that enhance cultural responsiveness in teacher–child interactions” would be more effective. It is uncertain whether such a narrowing is feasible within this current manuscript, but at present, the range the paper seeks to cover is far too broad.

Response 3.1: Thank you so much for your feedback. We acknowledge the importance focus on specific domains, but we argue that culturally responsive in social emotional learning is already very specific, given its unique focus on how teachers employ culturally responsive pedagogy in teaching social emotional learning skills for their students. Further, given the lack of frameworks in CR-SEL, it is extremely beneficial to have a broader scope to understand how teachers’ overall practice in CR-SEL.

  1. Position the theoretical framework more prominently
    It would be clearer if the Bennett framework or CASEL’s five core SEL competencies were explicitly presented at the beginning of the study and if the results were reorganized according to this framework. Currently, the connection between the theoretical framework and the themes is not clearly articulated, making the theoretical component appear weak.

Response 3.2: Thank you for pointing it out. We added that our study is grounded in Bennett’s framework of culturally responsive literacy practices in the introduction. We also reorganized our literature review by presenting the definition of CR-SEL and theoretical framework at the beginning of the study on page 2. To strengthen the connection between the framework and the theme, we explicitly explained in data analysis procedure how we combined the original framework’s dimensions to come up with our themes on page 6.  

  1. Clarify the theoretical contribution
    Although the study draws on the Bennett et al. (2018) framework, it should clearly indicate what new elements it adds to the framework or what modifications for rural contexts are suggested. Rather than simply applying the Bennett framework, the study should provide implications such as proposing a new model or concept for how CR-SEL should be modified or expanded in rural contexts.

Response 3.3: Thank you for your feedback. We revised our discussion to explicitly indicate the theoretical contribution on page 12.

  1. Present the activities (practical content) more concisely
    Reduce the number of interview quotations and instead present the core activities and their meaning through tables and diagrams. The cases should be structurally organized, and the main text should focus on summarized interpretations.
    Alternatively, specific activities or examples could be compiled into a table. Although the manuscript includes many teacher interview excerpts and examples, the activities are scattered throughout the narrative. Table 3 contains some activities, but they are very brief and at a keyword level. Even though the manuscript narratively organizes the activities under the “three major themes” (developing multicultural and critical perspectives, creating inclusive environments, and family engagement), they are not easily viewed at a glance.
    If a concrete list of activities were provided, the practical implications of the findings would be clearer and the usefulness to readers would increase. However, rather than simply listing activities, they should be organized by theme and connected to the theoretical context (CASEL’s five SEL competencies and the CR-SEL framework). We recommend summarizing them in a table or as an appendix rather than expanding the text. Indicating which SEL competencies each activity relates to would strengthen both the theoretical and practical contributions of the study.

Response 3.4: Thank you for your great feedback. We added a table in Appendix A including examples of quotations to elicit activities that teachers used for CR-SEL. We also substantially reduce the number of quotations to help with the flow.

  1. Emphasize structural issues in family engagement and language support
    The manuscript should stress that family engagement and language support are largely dependent on teachers’ individual capacities or personal networks. Framing this as a structural problem would give the study a stronger policy message.

Response 3.5: Thank you for your suggestion. We adapted your feedback by adding a statement that “However, most family engagement and language support were largely dependent on teachers’ individual capacities or personal networks.” on page 12.

Minor Comment

  1. Clarify the concept of TK
    Considering international readers, the concept of TK (Transitional Kindergarten) may be unfamiliar to those outside the United States. Adding one or two sentences to explicitly state that “TK is a transitional grade level between Kindergarten and Preschool” would make the manuscript clearer.

Response 3.6: Thank you for your suggestion. We added the statement on page 4 to make it clearer to our readers.

Round 2

Reviewer 2 Report

Comments and Suggestions for Authors

Dear authors,
I appreciate the important improvements you have made to the manuscript. I recommend the article for publication.
Congratulations and good luck in the future!

Best regards,